# Predicting Cyanobacterial Blooms Using Hyperspectral Images in a Regulated River

**DOI:** 10.3390/s21020530

**Published:** 2021-01-13

**Authors:** Jung Min Ahn, Byungik Kim, Jaehun Jong, Gibeom Nam, Lan Joo Park, Sanghyun Park, Taegu Kang, Jae-Kwan Lee, Jungwook Kim

**Affiliations:** 1Water Quality Assessment Research Division, Water Environment Research Department, National Institute of Environmental Research, Incheon 22689, Korea; ahnjm80@gmail.com (J.M.A.); kbi0102@korea.kr (B.K.); jongjaehun@korea.kr (J.J.); gbnam@korea.kr (G.N.); mintaka35@korea.kr (L.J.P.); pbaby75@korea.kr (S.P.); taegu98@korea.kr (T.K.); 2Water Environment Research Department, National Institute of Environmental Research, Incheon 22689, Korea; jkleenier@korea.kr

**Keywords:** water quality modeling, hyperspectral image, cyanobacterial bloom, Phytoplankton functional group, environmental fluid dynamics code

## Abstract

Process-based modeling for predicting harmful cyanobacteria is affected by a variety of factors, including the initial conditions, boundary conditions (tributary inflows and atmosphere), and mechanisms related to cyanobacteria growth and death. While the initial conditions do not significantly affect long-term predictions, the initial cyanobacterial distribution in water is particularly important for short-term predictions. Point-based observation data have typically been used for cyanobacteria prediction of initial conditions. These initial conditions are determined through the linear interpolation of point-based observation data and may differ from the actual cyanobacteria distribution. This study presents an optimal method of applying hyperspectral images to establish the Environmental Fluid Dynamics Code-National Institute of Environment Research (EFDC-NIER) model initial conditions. Utilizing hyperspectral images to determine the EFDC-NIER model initial conditions involves four steps that are performed sequentially and automated in MATLAB. The EFDC-NIER model is established using three grid resolution cases for the Changnyeong-Haman weir section of the Nakdong River Basin, where *Microcystis* dominates during the summer (July to September). The effects of grid resolution on (1) water quality modeling and (2) initial conditions determined using cumulative distribution functions are evaluated. Additionally, the differences in *Microcystis* values are compared when applying initial conditions using hyperspectral images and point-based evaluation data. Hyperspectral images allow detailed initial conditions to be applied in the EFDC-NIER model based on the plane-unit cyanobacterial information observed in grids, which can reduce uncertainties in water quality (cyanobacteria) modeling.

## 1. Introduction

In South Korea, four types of cyanobacteria, *Microcystis*, *Anabaena*, *Oscillatoria*, and *Aphanizomenon*, that produce trace amounts of odorous substances and toxins (microcystin, anatoxin, saxitoxin, etc.) have been designated harmful cyanobacteria and managed accordingly. Algae alert systems monitor the number of harmful cyanobacterial cells in water sources every seven days. The South Korean algae alert system divides alerts into four stages (below 1000 cells/mL is considered “normal,” between 1000 cells/mL and 10,000 cells/mL is the “advisory” level, between 10,000 cells/mL and 1,000,000 cells/mL indicates “caution,” and above 1,000,000 cells/mL is a “bloom”). Data acquisition on the growth and death of algae in grid units is not conducted for an entire river section. The location and composition of a large-scale algae bloom can alter in a short period due to environmental conditions such as light (sunshine), water temperature, nutrients (nitrogen and phosphorus), and residence time. There are temporal and spatial limits for algae management when only using algae data from water source areas. The algae concentration measured in a target area does not necessarily represent the distribution and concentration of algae for the entire region. Remote sensing can be used to address these problems by identifying a wide range of algae bloom conditions at a resolution not available with field measurements [1]. Recently, various remote sensing techniques have been studied using unmanned aerial vehicles (UAVs) [2,3,4,5]. In addition, chlorophyll-a (Chl-a), which corresponds to algal blooms, has also been monitored using UAVs [4,6].

Both domestic and foreign studies have utilized hyperspectral images in algae forecasts. Choi et al. [7] estimated the Chl-a concentration in the Nakdong River Basin in South Korea using high-resolution satellite images. Park et al. [8] reviewed and analyzed studies on the application of hyperspectral sensors in monitoring water quality, particularly for phytoplankton. Kim et al. [9] used UAVs to capture aerial images of the Dodong pier in the middle of the Nakdong River and to derive an exponential formula for detecting algae. This exponential formula was highly correlated with the phytoplankton quantity, and it demonstrated the potential applications of algae monitoring using UAVs. Recently, a study monitored algae using remote sensing data from the Landsat 8 satellite. Lim et al. [10] estimated the total nitrogen and total phosphorous concentrations in the Geumgang River basin using image data from the Landsat 8 satellite. The study monitored the occurrence of algae by comparing and verifying the total nitrogen and total phosphorous concentrations using multiple linear regression formulas. Jang et al. [11] quantified the nutritional status of Jinyang Lake by analyzing its Chl-a concentration spatial distribution using Landsat 8 images. Satellite remote sensing data have also been used to monitor water quality items related to algae concentration or blooms. Zhang et al. [12] used satellite remote sensing technology to monitor the Chl-a trend, and Adam [13] developed an empirical remote sensing model to estimate Chl-a and harmful cyanobacteria. Hansen et al. [14] utilized remote sensing technology to predict algal growth using Landsat data. Additionally, Ortiz et al. [15] addressed issues related to atmospheric correction, noise reduction, and mixed hyperspectral image pixels using composition analysis. Sawtell et al. [16] monitored the behavior of harmful cyanobacteria in real-time based on high-resolution images by performing noise reduction and atmospheric correction on National Aeronautics and Space Administration hyperspectral images. Woude et al. [17] collected hyperspectral images to monitor the propagation of harmful cyanobacteria in the United States Great Lakes. The temporal and spatial variations of harmful cyanobacteria were then analyzed using the collected data. Thus, the occurrence and behavior of algae have been monitored in real-time using both point and plane units and hyperspectral images.

Most water quality-monitoring studies have used satellite data and hyperspectral images to predict the algal behavior or monitor water quality items that cause algae to bloom. In other fields, remote sensing data are not only used for monitoring but are also integrated into models for data analysis, including that for evapotranspiration estimation [18], atmospheric wind prediction [19], and ground surface radiation and energy estimation [20] in the meteorological field and flood risk assessment in the flood disaster field. Recently, a study was conducted in the water quality field to predict pollution in Donghu Lake by applying remote sensing data to the MIKE 21 and multi-source nonlinear regression fitting models [21]. There are few studies in which remote sensing data are directly applied in numerical models to predict harmful algae. This is because modeling living organisms, such as algae, involves many uncertainties. Therefore, remote sensing data from ungauged areas can be utilized to confirm and validate modeling results [22,23]. However, remote sensing data have not yet been directly applied to model future algal blooms.

The National Institute of Environmental Research (NIER) acquires hyperspectral images using UAVs to observe Chl-a and phycocyanin concentrations [24,25,26,27,28]. Recently, the occurrence and behavior of algae have been accurately monitored in real-time using remote sensing data. However, to identify and manage water environment problems, such as repeated summertime algal blooms, it is necessary to predict changes in short-term algae concentration using water quality prediction models. The initial condition of algae distribution in water is particularly important for the prediction of short-term algae concentration because it affects the accuracy of the prediction result. Thus, the use of an accurate initial condition of algae can reduce uncertainties in the prediction results. Hyperspectral images encompassing the actual algae concentration values across the entire section can be used to determine accurate algae initial conditions.

The purpose of this study is to generate the initial conditions for the current water quality prediction model Environmental Fluid Dynamics Code (EFDC)-NIER using hyperspectral image data and evaluate its applicability in short-term algae forecasts. The optimal method for applying hyperspectral image data in EFDC-NIER grids and the optimal grid resolution of the EFDC-NIER model to predict algae are presented. The study was conducted in the following steps: (1) an EFDC-NIER model was constructed for the Changnyeong-Haman weir section with a dominant algae presence, (2) the representative Chl-a concentration was calculated to apply hyperspectral images in determining the EFDC-NIER initial condition, (3) the prediction sensitivity of cyanobacteria based on the calculated Chl-a concentration and the predictive powers of three grid resolutions (5 partitions, 10 partitions, and 20 partitions) were compared and analyzed, (4) the optimal grid resolution for applying hyperspectral images in EFDC-NIER was presented, and (5) the applicability of the hyperspectral image-based initial condition for the water quality model was evaluated.

## 2. Materials and Methods

### 2.1. EFDC-NIER

The EFDC model is a three-dimensional numerical model developed by the Virginia Institute of Marine Science in the early 1990s; it has since been managed and supplemented by the U.S. Environmental Protection Agency. The EFDC model is widely used worldwide to understand hydraulic and water quality behaviors in various areas, including rivers, lakes, estuaries, and seas. Since 2010, the National Institute of Environmental Research has improved the function of the EFDC (20100328 version) source code to suit the conditions of major waters in South Korea and has developed the necessary modules to officially use the model as a water quality forecast model for major sections of South Korean rivers. The improved model was named EFDC-NIER. The EFDC-NIER model has been equipped with new features, such as incorporating the weir function of major rivers in South Korea, multi-species algae simulation, the vertical movement mechanism of cyanobacteria, dormant spore generation and germination, wind stress, and bottom-water nutrient elution variations due to changes in oxidation and reduction conditions (Figure 1).

Since the Four Major Rivers Project, the EFDC-NIER model has been improved to more accurately reflect changes in flow rate and water level due to artificial hydraulic structures, such as multifunctional weirs, thereby improving the simulation accuracy of the changed river environments. In particular, the existing EFDC model simulates algae by classifying them into three different species (cyanobacteria, diatoms, and other algae), making it difficult to predict the rapid dominance and transitions of certain algae. However, the EFDC-NIER model can be utilized to quantitatively predict the occurrence of algae, including their rapid dominance and transitions because the algae module has been enhanced to allow for multi-species simulation (Figure 2).

### 2.2. Hyperspectral Image Application Method in EFDC-NIER Model 

In this study, hyperspectral images were taken of the Changnyeong-Haman weir section of the Nakdong River Basin, and the algae monitoring data observed using this remote sensing technique were applied to determine the initial condition of the EFDC-NIER model (Figure 3).

The hyperspectral images were acquired using the AISA Eagle sensor mounted on a UAV. The acquired images were radiometrically and geometrically corrected using Caligeo Pro, and an atmospheric correction was performed using ATCOR-4. The spectral data of the water measured on-site on the day of filming, phycocyanin pigment concentration, and cyanobacteria cell count from the same location were used to obtain the cyanobacterial information of the hyperspectral image data [27]. First, the genetic algorithm method was used to estimate the phycocyanin pigment concentration based on the spectral data, and the R^2^ value (0.85) of the learning and verification data indicated strong explanatory power. The cyanobacterial cell count was derived from the phycocyanin concentration and linear regression analysis. The R^2^ value of the regression equation was 0.71, indicating strong explanatory power. In this study, the cyanobacteria cell count distribution generated through the process described earlier was applied to the initial condition of the EFDC model.

There are various modeling input conditions, such as weather, boundary, initial, and hydraulic structure operation conditions, for modeling *Microcystis* in the EFDC-NIER model. Of these, the initial condition acts as an important element in the short-term forecast for *Microcystis*. The initial condition was applied in the model based on the linear interpolation of the observed point-to-point data. Using the Chl-a values obtained at the observation points across the water quality monitoring network, the EFDC-NIER model grids were interpolated using the nearest neighbor interpolation method and then applied to the initial condition for modeling. When the distance between the monitoring network points is large, the initial distribution of algae cannot be accurately reflected. In contrast, as grid-format data observed through aerial imaging, hyperspectral images can provide an initial algae distribution that is similar to that in reality. The procedure for applying hyperspectral images in the EFDC-NIER model is displayed in Figure 4. The steps were automated using MATLAB.

The first step is to extract the Chl-a concentration value of each grid in the hyperspectral image and group the corresponding data using the EFDC-NIER model grid. Various Chl-a concentration values from the hyperspectral image are entered into each EFDC-NIER model grid, depending on the difference in the spatial resolution between the two datasets (Figure 5).

The second step is to interpolate the no data that contain no hyperspectral Chl-a concentration values due to hydraulic structures, such as bridges and weirs, using the average concentration value of the adjacent EFDC-NIER grids. The third step is to calculate the representative Chl-a concentration value of the hyperspectral image data grouped into EFDC-NIER model grids. It is difficult to calculate the representative concentration value because the concentration value distribution varies for each grid. Thus, to calculate the representative Chl-a concentration values, the cumulative distribution function (CDF) and mean, representing the appropriate properties of the set of values, were applied. For each grid, the Chl-a concentration values were calculated using CDFs in the first quartile (25%), middle quartile (50%), and third quartile (75%). Additionally, the mean Chl-a concentration value for each grid was arithmetically determined. As the final step, the initial algae field file (WQWCRSTX.inp) was generated by converting the representative Chl-a value of each EFDC-NIER model grid to carbon according to the carbon ratio of each Phytoplankton Functional Group (PFG) codon.

To apply the generated initial condition of the hyperspectral data to the EFDC-NIER model, nine representative PFG codons were selected based on the algal species listed by Reynolds et al. [29] and Padisak et al. [30], and those observed in the Nakdong River Basin (Table A1, Appendix A). Figure 6 displays the 684 algal species found 500 m upstream of the Changnyeong-Haman weir in the Nakdong River Basin grouped by PFG codon and the cell count by codon observed between 7 January 2019, and 23 December 2019. The biovolume value per unit cell in Figure 6 was calculated using the average cell length, width, and thickness of each algal species found in the Nakdong River in 2016, as listed in Table A2 (Appendix A) [25]. The carbon content per codon was calculated by converting the cell count of the observed algal species based on the pgC/cell for each species provided in Table A2 (Appendix A). For example, given a cell count of 10,000 cells/mL for *Microcystis* spp., the carbon content is 10,000 × 10.95/1,000,000 = 0.1095 mg C/L. Based on the cell counts of the 684 algal species observed in the water quality monitoring network 500 m upstream of the Changnyeong-Haman weir, the monthly carbon ratio was calculated for each of the nine PFG codons (Figure 6). The monthly carbon ratio defines the carbon content of each codon in the tributary inflow as the boundary condition in the EFDC-NIER model. The monthly carbon ratio for the initial condition was calculated using the same method, based on the cell count for each codon observed on the same day as when the hyperspectral image was taken 500 m upstream of the Changnyeong-Haman weir. The carbon ratio was calculated for each PFG codon by measuring the cell count of the 684 algal species found 500 m upstream of the Changnyeong-Haman weir on 6 July 2019. The harmful cyanobacteria, *Microcystis* (Codon M), resulted in 9.5%.

The carbon–Chl-a ratio (β) is required to convert the Chl-a value into carbon. For the eight observation points in the Nakdong River Basin (Sangju weir, Nakdan weir, Gumi weir, Chilgok weir, Gangjeong-Goryeong weir, Dalseong weir, Hapcheon-Changnyeong weir, Changnyeong weir, Changnyeong weir, and Changnyeong-Haman weir), the ratio of the carbon contents per codon to the Chl-a values observed between 2013 and 2018 was 0.12 on average. Therefore, a β of 0.12 was applied in this study. The equation for converting the representative Chl-a value of each EFDC-NIER model grid extracted from the hyperspectral image to the carbon content by codon is provided below:(1)$$\mathrm{Codon}X_{all}carbon=\;\sum_{i=1}^{9}(\beta \times CodonX_{i}carbon\;ratio\;\times Chl-a\;concentration)$$
where *i* represents the codon (M, H1, D, C, X2, P, G, J, and LO), and *β* is the *carbon*–*Chl*-*a concentration* ratio (0.12).

The results of applying the calculated carbon content of Codon M in the initial condition of the EFDC-NIER model using both the hyperspectral image (grid unit) and monitoring network data (point unit) are illustrated in Figure 7. The initial condition generated with the monitoring network data simplified the initial field because the ungauged grids were linearly interpolated using the point-to-point observation data from 500 m upstream of each multifunctional weir. In contrast, the initial condition generated using the hyperspectral image reflected the actual algae occurrences because the Chl-a concentration values were measured for all grids (Figure 7).

### 2.3. Study Area and Model Construction 

In this study, the Changnyeong-Haman weir section of the Nakdong River Basin was selected as the target area to assess the applicability of hyperspectral images in water quality (algae) prediction (Figure 8).

The Nakdong River Basin is an area with frequent summertime algal blooms due to its topographical features. In particular, the Changnyeong-Haman weir section, located in the downstream area of the Nakdong River Basin, serves as a water source, making it a suitable section for the study. Factors affecting the water supply, such as tributaries flowing into the primary river stream, sewage plant discharges, and water intake stations, were reflected in the model as boundary conditions. A “mask” was set on the grids where multifunctional weirs were located, and the upstream inflows were set to be discharged downstream using a hydraulic structure module. Municipal meteorological observation data from the Korea Meteorological Administration open weather data portal, daily operation data provided by K-water for Changnyeong-Haman weir water level management, daily dam discharge data from the Water Resources Management Information System, flow rate observation data from the Ministry of Environment, and water quality monitoring network data from the Ministry of Environment were used in the study.

The Chl-a values in the 2 × 2 m lattices of the hyperspectral image were matched to the EFDC-NIER model grids and grouped to calculate the representative Chl-a values. Depending on the grid resolution of the EFDC-NIER model, the Chl-a value varied for each EFDC-NIER model grid. The EFDC-NIER model with the same resolution as the 2 × 2 m grid in the hyperspectral image did not require a representative Chl-a value calculation, and more grids in the EFDC-NIER model resulted in a longer modeling time. The optimal grid resolution for predicting water quality (algae) while achieving an efficient calculation was evaluated in this study. There were three cases based on the number of horizontal grids. To build the EFDC-NIER model with various study area resolutions, different numbers of grids were set for the stream longitudinal (I-direction) and latitudinal (J-direction) directions. In Case 3, grids were constructed within a range that ensured orthogonality with 20 grid partitions in the J-direction (Figure 9c). Based on Case 3, the I- and J-direction grids in Case 2 were divided in half (10 partitions, Figure 9b), and the I- and J-direction grids were divided in half again (5 partitions) in Case 1 (Figure 9a). The total numbers of horizontal grids were 1105 for Case 1, 4430 for Case 2, and 17,700 for Case 3. The vertical direction (K-direction) grid was set in five partitions for all three cases.

## 3. Results and Discussion

### 3.1. Long-Term Water Quality Sensitivity Analysis by Grid Resolution 

In this study, the EFDC-NIER model was constructed with three different grid resolutions. The effect of grid resolution on water quality (algae) prediction was first evaluated. Parameter correction was performed on the water quality items and algae-related water quality based on Case 3, which exhibited the highest grid resolution. The discharge water from the Hapcheon-Changnyeong weir was applied as the upstream boundary, and the water level at the Samrangjin Water Level Monitoring Station was applied as the downstream boundary. Within this constructed section of the model, the Changnyeong-Haman weir was reflected as a hydraulic structure. The evaluation period was set from 20 June 2019, to 10 August 2019, and a comparison with the observed values was conducted based on a point located 500 m upstream of the Changnyeong-Haman weir.

Model parameter correction was conducted using the mean absolute error (MAE) and root mean square error (RMSE). MAE is the average of the absolute error of the observed and simulated values, and it can be used to compare the residuals between the models, while RMSE is the average error of the observed and simulated values which indicates the model precision. The values listed in Table 1 were used as the primary parameters of the algae and water quality analyses. The RMSE and MAE analysis results for the simulation period are provided in Table 2. Based on the results, the model reasonably simulated the changing water flow characteristics (water level and water temperature) in the weir section, which fundamentally dictates algal blooms and behavior patterns, and water quality items (nutrients and organics) that are highly correlated to algal blooms and behavior patterns.
(2)$$\mathrm{MAE}=\frac{\sum_{i=1}^{N}\left|O_{i}-P_{i}\right|}{N}$$
(3)$$\mathrm{RMSE}=\sqrt{\frac{1}{N}\sum_{i=1}^{N}{\left(O_{i}-P_{i}\right)}^{2}}$$
where *P_i_* is the simulated value at time *i*, *O_i_* is the observed value at time *i*, and *N* is the number of observed values for the entire period.

Figure 10 displays a graphical representation of the analysis results and observed values of biochemical oxygen demand (BOD), total nitrogen (TN), total phosphorus (TP), Chl-a, and cyanobacterial cell counts simulated 500 m upstream of the Changnyeong-Haman weir. Under identical environmental conditions, including weather, boundary, initial, and hydraulic structure operation conditions, changes in grid resolution did not appear to affect the water quality model results. Thus, the sensitivity of the water quality model to grid resolution can be considered small for a one-dimensional time series analysis. In one-dimensional time series modeling, a multidimensional model is no more accurate than a one-dimensional numerical model. The increased grid resolution in a multidimensional model can disrupt fast decision-making because it requires a longer simulation time. Therefore, a one-dimensional model or a low-resolution multidimensional model is deemed sufficient when decisions must be made quickly.

### 3.2. Applicability of EFDC-NIER Initial Condition Based on Representative Concentration Value and Grid Resolution 

Figure 11 displays the *Microcystis* cell count results 500 m upstream of the Changnyeong-Haman weir in the short-term, with three-day predictions performed by applying the hyperspectral image data taken on 2 August 2018, to the initial condition for each case and CDF. If the EFDC-NIER model was built with the same grid as the 2 m × 2 m grid resolution of the hyperspectral image, the average and CDF (25%, 50%, and 75% quartiles) achieve the same modeling results. As illustrated in Figure 11, the difference between the CDF 75% and 25% quartile models was the smallest in Case 3. The spread between the two values increased in the order of Case 3 < Case 2 < Case 1. The higher the grid resolution of the EFDC-NIER model, the smaller the short-term *Microcystis* forecast deviation was when applying the representative Chl-a value from the hyperspectral image, resulting in identical CDF 50% quartile and average values. High water temperatures of 30.7 °C and 31.8 °C were observed on 30 July 2018, and 6 August 2018, respectively, providing the optimal conditions for *Microcystis* to grow. A representative EFDC-NIER grid value should represent the respective grid value. The lowest values were typically found with CDF 50%, with no significant difference in the simulation results when the CDF 50% and average values were used. Therefore, it was deemed appropriate to calculate the representative Chl-a values using the median CDF 50% values from the hyperspectral images of the Changnyeong-Haman weir section.

Figure 12 displays the short-term prediction results of the *Microcystis* cell count after applying the initial condition of 50% CDF to each grid resolution case using Chl-a data from the hyperspectral image observed on 6 July 2019. The results correspond to a 7 days modeling period from 6 July 2019, to 13 July 2019, which demonstrated that the higher the grid resolution, the more accurately the algal distribution was expressed. Case 3, which exhibited a small spread and precise distribution of algae, is considered the optimal grid resolution because the optimal method for calculating representative concentrations can vary depending on the environmental conditions. In particular, the maximum algae bloom in the dead water zone on the right riverbank near a river island was predicted to be 22,009 cells/mL in Case 2 and 21,735 cells/mL in Case 3, which were higher than the 16,164 cells/mL prediction in Case 1.

### 3.3. Hyperspectral Image Applicability in Water Quality Model Initial Field

Figure 13 displays the result of modeling *Microcystis* cell count from 6 July 2019, to 22 July 2019, by applying the Chl-a values from the hyperspectral image (HSI) taken on 6 July 2019, and the carbon content by the PFG codon measured at the monitoring network in the initial condition of the EFDC-NIER model. While the effect of the initial condition used in the two methods varied for the first seven days, from 6 July 2019, to 13 July 2019, the *Microcystis* cell count modeling results were still similar, and the effect of the initial conditions was no longer noticeable (Figure 13). Different results were observed depending on how the initial conditions were applied. The water temperature was 24 °C on 1 July 2019; 26.2 °C on 8 July 2019; and 25.6 °C on 15 July 2019, a low water temperature compared to the average of 31.4 °C between 30 July 2018, and 6 August 2018. This result indicated that the effect of the initial condition based on CDF was small. The application of the initial conditions calculated by linearly interpolating the algal data between the observation points of the monitoring network in the EFDC-NIER model can either over- or under-apply the carbon content. In contrast, the use of hyperspectral images allows detailed initial conditions to be applied in the EFDC-NIER model based on plane-unit algae data measured in grids, reducing uncertainties in water quality modeling (Figure 14).

## 4. Conclusions

This study presented the optimal method for applying hyperspectral images as a remote sensing technique in the initial condition for the short-term prediction of *Microcystis*. (1) A sensitivity analysis of the water quality simulation for different EFDC-NIER model grid resolutions, (2) a comparison of results at different grid resolutions and data resampling between the hyperspectral image and EFDC-NIER model, and (3) a comparison of results between the existing point-based initial conditions from the data observed at the monitoring network and the grid-unit hyperspectral image-based initial condition were performed in this study. The major findings of this study are as follows.
The sensitivity of the water quality simulation was small for varying initial conditions, boundary conditions, and parameters. In a one-dimensional time series analysis, a multidimensional model is no more accurate than a one-dimensional numerical model, even at a higher grid resolution. While a multidimensional model is necessary when modeling a dead water zone that requires high spatial accuracy, a low-resolution model is deemed sufficient for quick decision-making and conducting a one-dimensional time series analysis. It is critical to select and operate a model that is appropriate for the purpose and circumstances.When resampling different grid resolutions between the hyperspectral image and EFDC-NIER model, the dispersion of the results with different CDFs decreased as the EFDC-NIER model grid resolution increased. Case 3 is the most optimal grid resolution, and CDF 50% should be used to reduce the effect of various environmental conditions on the modeling result.When using linearly interpolated algae data from the observation points across the monitoring network, the carbon content may be under- or over-applied. The use of hyperspectral images can reduce uncertainties in the modeling results because detailed initial conditions can be applied to the target section.As various remote sensing techniques, such as satellite images, are being studied in addition to hyperspectral images, if the Chl-a or algal cell count data can be directly observed and provided, these data can be used in the initial condition of hydrodynamic models using the method presented in this study.

## Figures and Tables

**Figure 1 sensors-21-00530-f001:**
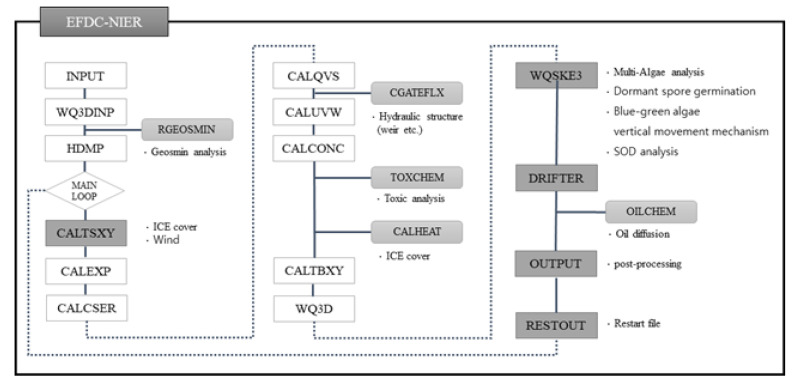
Environmental Fluid Dynamics Code-National Institute of Environment Research (EFDC-NIER) schematic.

**Figure 2 sensors-21-00530-f002:**
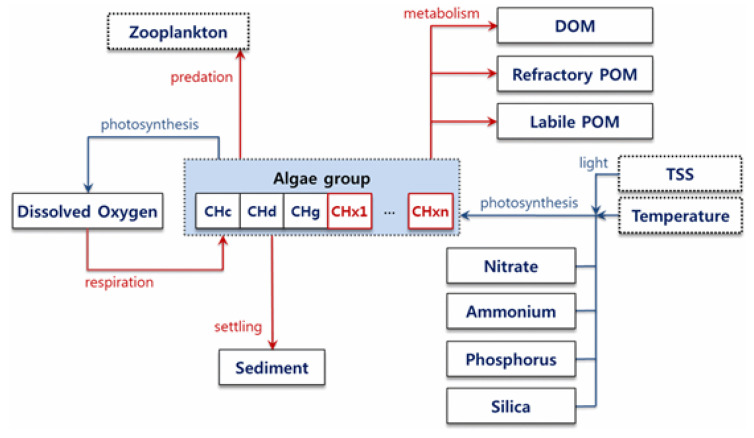
Multi-species algae simulation module schematic.

**Figure 3 sensors-21-00530-f003:**
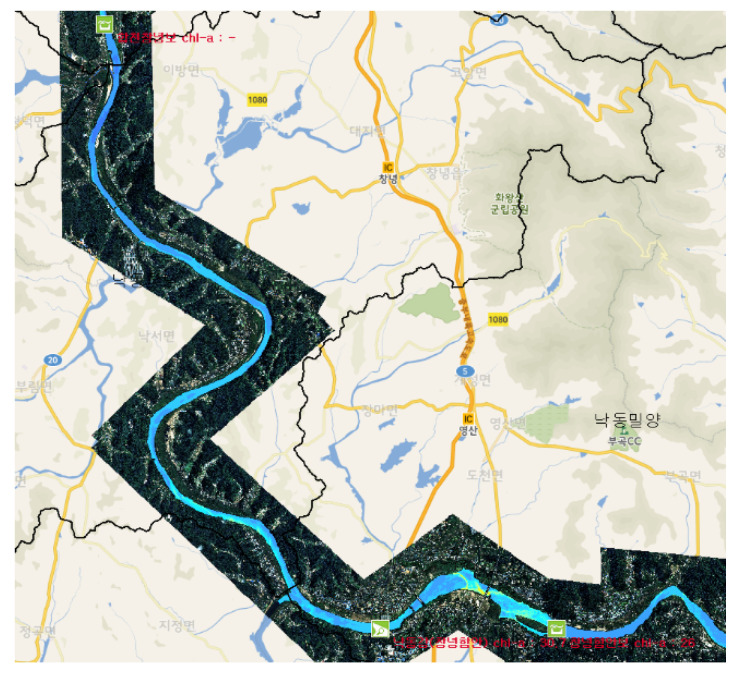
Algal distribution hyperspectral remote sensing using inherent optical properties [28].

**Figure 4 sensors-21-00530-f004:**
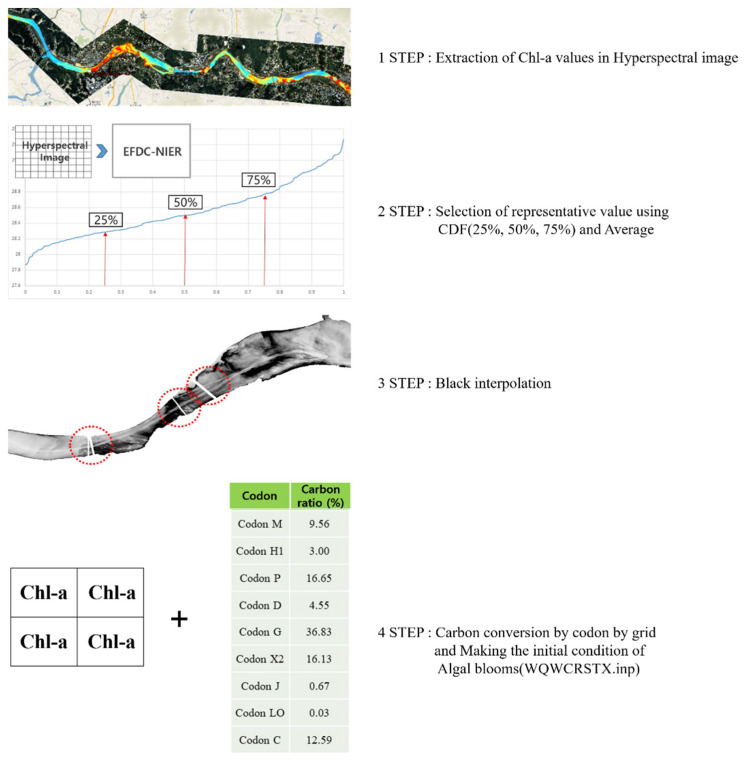
Procedure for applying hyperspectral remote sensing data in initial field of EFDC-NIER model.

**Figure 5 sensors-21-00530-f005:**
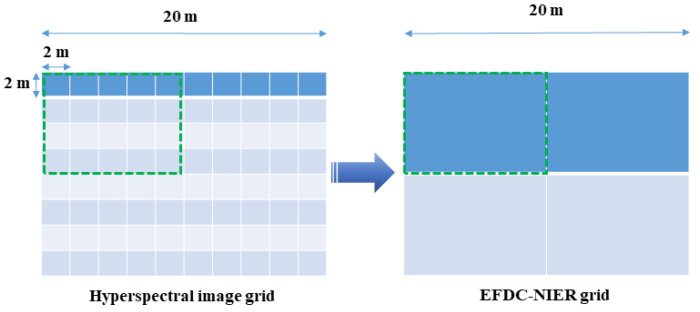
Resampling from the hyperspectral image grid to the EFDC-NIER model grid.

**Figure 6 sensors-21-00530-f006:**
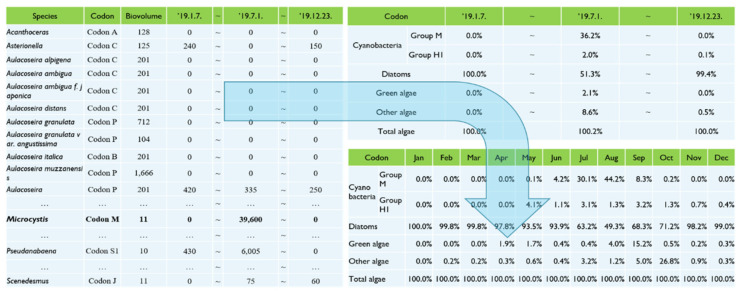
Calculation of carbon ratio for each Phytoplankton Functional Group (PFG).

**Figure 7 sensors-21-00530-f007:**
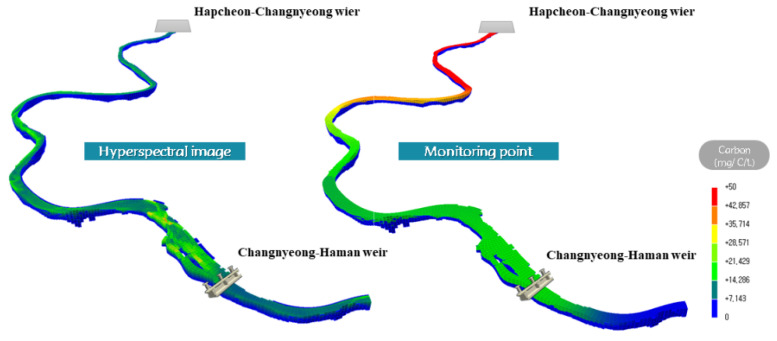
Initial condition application for *Microcystis* using hyperspectral and monitoring point network data.

**Figure 8 sensors-21-00530-f008:**
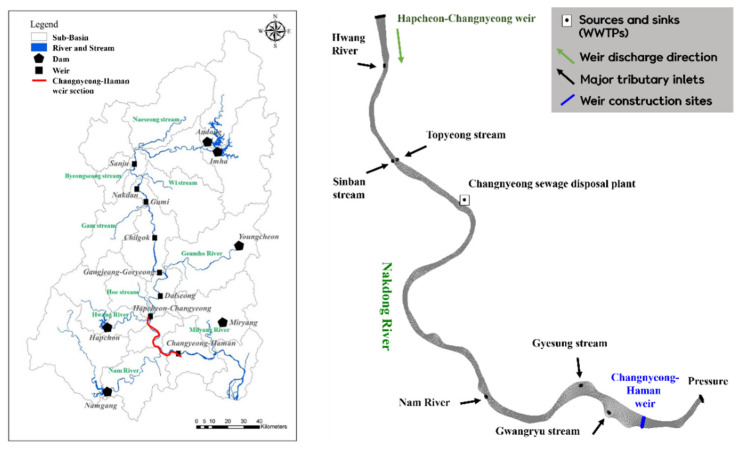
Study area (Left: Nakdong River Basin; Right: Changnyeong-Haman weir section).

**Figure 9 sensors-21-00530-f009:**
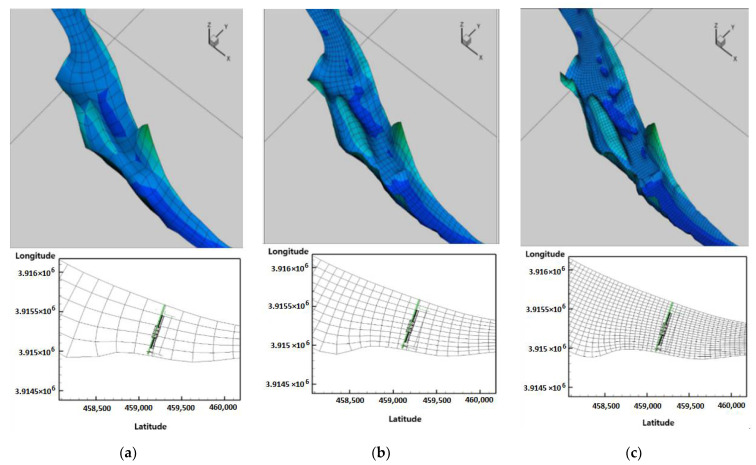
Model construction for each grid resolution: (**a**) Grid resolution of case 1; (**b**) Grid resolution of case 2; (**c**) Grid resolution of case 3.

**Figure 10 sensors-21-00530-f010:**
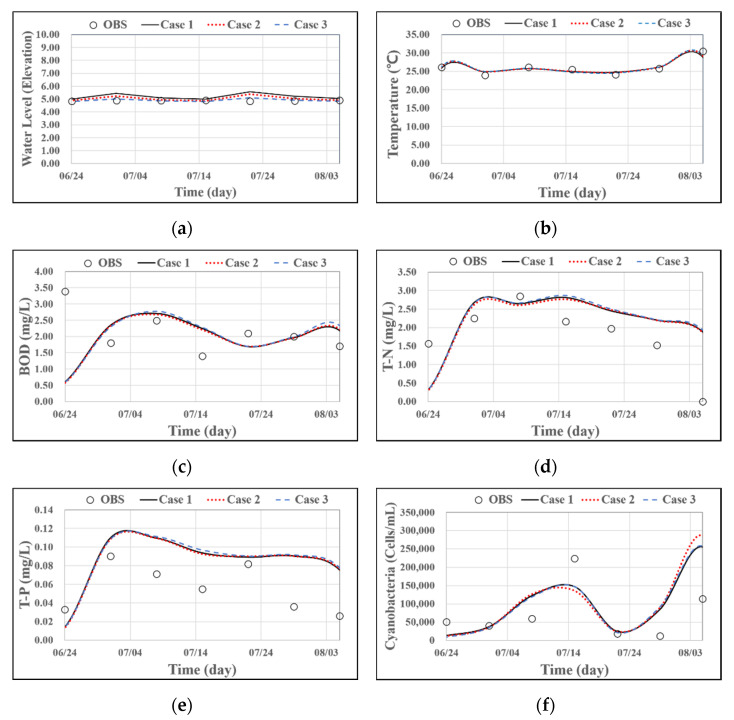
Sensitivity of long-term water quality parameters by case: (**a**) water level; (**b**) temperature; (**c**) BOD; (**d**) T-N; (**e**) T-P; (**f**) cyanobacteria.

**Figure 11 sensors-21-00530-f011:**
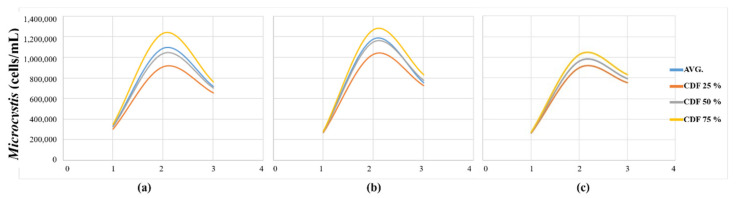
Results of applying hyperspectral image-based initial conditions for representative Chl-a concentration estimations: (**a**) *microcystis* modeling results in Case 1; (**b**) *microcystis* modeling results in Case 2; (**c**) *microcystis* modeling results in Case 3.

**Figure 12 sensors-21-00530-f012:**
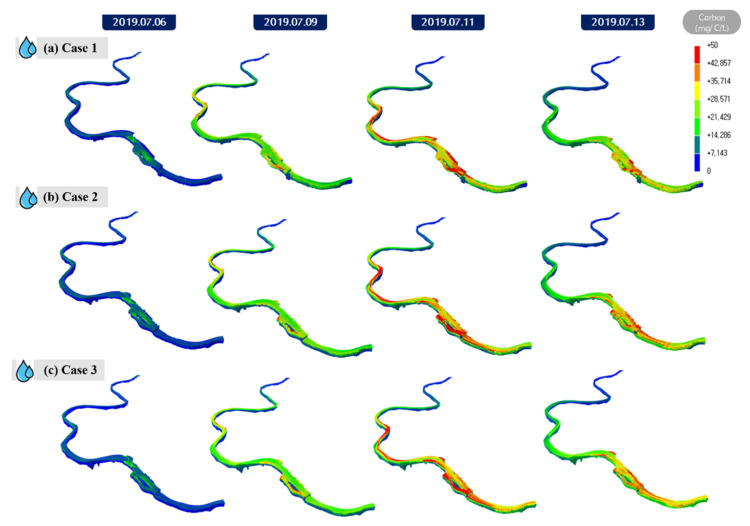
Results of applying cumulative distribution function (CDF) 50% to the initial field in each case: (**a**) carbon concentration of hyperspectral image in Case 1; (**b**) carbon concentration of hyperspectral image in Case 2; (**c**) carbon concentration of hyperspectral image in Case 3.

**Figure 13 sensors-21-00530-f013:**
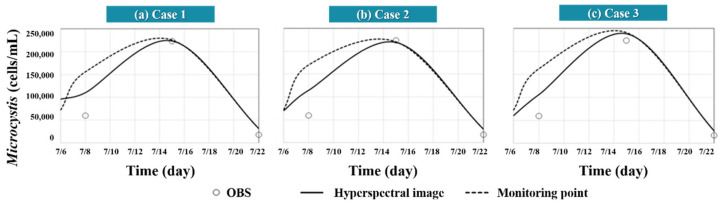
Comparison of initial field application method modeling results in each case. (**a**) microcystis modeling results using HSI and Monitoring data in Case 1; (**b**) microcystis modeling results using HSI and Monitoring data in Case 2; (**c**) microcystis modeling results using HSI and Monitoring data in Case 3.

**Figure 14 sensors-21-00530-f014:**
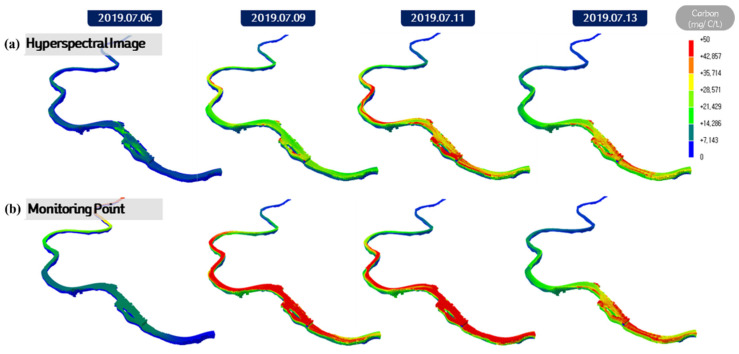
Comparison of initial condition monitoring results based on hyperspectral and monitoring network data: (**a**) carbon concentration of hyperspectral image; (**b**) carbon concentration of monitoring point.

**Table 1 sensors-21-00530-t001:** Major parameters and ranges.

EFDC Parameter *		Unit	Definition	Nakdong River
PM_x_	Codon M	d	Maximum Growth Rate	3.0–4.0
	Codon H1			0.2–3.0
	Codon P			1.3–3.0
	Codon D			3.0–4.0
	Codon G			0.8–1.8
	Codon X2			1.5–3.5
	Codon J			1.2–1.5
	Codon LO			0.2–2.0
	Codon C			1.0–3.5
KHN_x_	Codon M	mg/L	Nitrogen Half-Saturation	0.03
	Codon H1			0.03
	Codon P			0.07
	Codon D			0.07
	Codon G			0.05
	Codon X2			0.05
	Codon J			0.05
	Codon LO			0.05
	Codon C			0.07
KHP_x_	Codon M	mg/L	Phosphorus Half-Saturation	0.01
	Codon H1			0.02
	Codon P			0.01
	Codon D			0.01
	Codon G			0.01
	Codon X2			0.01
	Codon J			0.01
	Codon LO			0.01
	Codon C			0.01
TMX_1_	Codon M	°C	Lower Optimal Temperature	20.0
	Codon H1			10.0
	Codon P			5.0
	Codon D			2.0
	Codon G			20.0
	Codon X2			2.0
	Codon J			18.0
	Codon LO			10.0
	Codon C			5.0
TMX_2_	Codon M	°C	Upper Optimal Temperature	35.0
	Codon H1			35.0
	Codon P			35.0
	Codon D			13.0
	Codon G			35.0
	Codon X2			30.0
	Codon J			32.0
	Codon LO			30.0
	Codon C			30.0
WQRHOMN	Codon M	kg/m^3^	Algae Minimum Density	985
	Codon H1			920
	Codon G			970
	Codon LO			920
WQRHOMX	Codon M	kg/ m^3^	Algae Maximum Density	1,005
	Codon H1			1,030
	Codon G			1,065
	Codon Lo			1,030
WQCOEF1	Codon M	kg/ m^3^/min	Density Increase Rate Constant	0.030
	Codon H1			0.070
	Codon G			0.045
	Codon Lo			0.070
WQCOEF2	Codon M	kg/ m^3^/min	Density Decrease Rate Constant	0.001
	Codon H1			0.001
	Codon G			0.001
	Codon Lo			0.001
WQCOEF3	Codon M	kg/ m^3^/min	Density Increase Minimum Rate	0.013
	Codon H1			0.023
	Codon G			0.011
	Codon Lo			0.023
WQR	Codon M	m	Algae Effective Radius	0.00008
	Codon H1			0.000005
	Codon G			0.00025
	Codon Lo			0.00002
CChl_x_		mg C/μg Chl-a	Carbon–Chl-a Ratio for Algae	0.012
CIa, CIb, Clc		-	Weighting Factor for Solar Radiation at 0 d, 1 d, and 2 d	0.80, 0.15, and 0.05
BMR_x_		/d	Basal Metabolism Rate for Algae	0.05–0.1
PRR_x_		/d	Predation Rate for Algae	0.02
CP_prm1_		g C/g P	Constant for Algae Phosphorous–Carbon Ratio	40
CP_prm2_		g C/g P	Constant for Algae Phosphorous–Carbon Ratio	85
CP_prm3_		mg/L	Constant for Algae Phosphorous–Carbon Ratio	200
ANC_x_		g N/g	Nitrogen–Carbon Ratio for Algae	0.18
L_Factor1		W/m^2^	Conver Light Unit	4.57
F_PAR			Temperature and Light Average Time	0.44

* Subscript c, d, g, and x indicate cyanobacteria, diatom, green algae, and algae, respectively.

**Table 2 sensors-21-00530-t002:** Model performance based on parameter correction results.

Group	Water Level (m)	Water Temperature (°C)	BOD (mg/L)	TN (mg/L)	TP (mg/L)
MAE	0.11	0.54	0.54	0.55	0.04
RMSE	0.15	0.69	0.62	0.61	0.04

## Data Availability

The data presented in this study are available on request from the corresponding author.

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
