# Peer review of "Predicting Cyanobacterial Blooms Using Hyperspectral Images in a Regulated River"

_sensors, 2021, doi:10.3390/s21020530_

Round 1

Reviewer 1 Report

The blue pigment, phycocyanin is found in blue-green algae and cyanobacteria, which is specific pigment for their identification. The Chl-a concentration is typical for to other algalgoups (diatoms, green algae and euglenoid algae).  Based on spectral analysis the Chlorphyll-a not suitable for identifying the amount of cyanobacteria. The available instruments are already able to measure cyanobacteria and other algae specifically due to differences in their spectral composition (https://www.bbe-moldaenke.de/en/products/chlorophyll/details/PhycoLabAnalyser.html). In summary, the change in the amount of cyanobacteria cannot be estimated from chlorophyll-a, please use the phycocyanin.

Cyanobacteria occur not only in the upper layers of water but also in the lower layers. Especially when the water becomes slower flowing. In such cases, I believe that the chosen method is not able to register the occurrence of living organisms in the deeper parts of the water.

Author Response

Thank you for your review. Plaease see the attachment.

Reviewer 2 Report

The remote methods of water quality assessment are developing rapidly. They ensure quite fast analysis of actual water state, including phytoplankton blooms, but they also provide tools for the predictions of future changes. The manuscript presents another possibility how to use the hyperspectral images to predict the bloom formation in time in river section. I find it highly useful in water quality monitoring. 

Used model was based on e.g. the list of species noted in studied river, however I have not found any information about the source of this data - how and where were the phytoplankton community analyzed, with what methods. Were these results published?

I think that the part of the figures is not cler enough - the font of the text is too small or the quality of the graph is too low. Moreover, most of the figures were not numbered. This must be improved as it influences on the results presentation.
